# Use of multifocal electroretinograms to determine stage of glaucoma

**Naoya Moroto**[1,2]*, **Shunsuke Nakakura**[3], **Hitoshi Tabuchi**[3,4], **Kiyofumi Mochizuki**[1], **Yusuke Manabe**[1], **Hirokazu Sakaguchi**[1]

1 Department of Ophthalmology, Gifu University Graduate School of Medicine, Gifu, Japan,
2 Ophthalmology, Ogaki Municipal Hospital, Ogaki, Japan, 3 Ophthalmology, Saneikai Tsukazaki Hospital, Himeji, Japan, 4 Department of Technology and Design Thinking for Medicine, Hiroshima University, Hiroshima, Japan

* naoya_5_3@hotmail.com

**Data Availability Statement:** All relevant data are within the manuscript and its Supporting Information files.

**Funding:** The authors received no specific funding for this work.

## Abstract

### Purpose

To determine whether multifocal electroretinograms (mfERGs) recorded with natural pupils and skin electrodes can be used to determine the stage of open angle glaucoma (OAG).

### Methods

Two hundred eighteen eyes of 132 OAG patients and 62 eyes of 62 normal subjects whose best-corrected visual acuity (BCVA) was 0.1 logarithm of the minimum angle of resolution (logMAR) units (20/25) or less were studied. The mean deviations (MDs) obtained by Humphrey Visual Field Analyzer (HFA), optical coherence tomographic (OCT) images, and mfERGs were analyzed. The glaucoma was classified into 4 stages: preperimetric glaucoma (PPG), early stage, moderate stage, and advanced stage glaucoma. The parameters of the mfERGs examined were the amplitudes of the two positive peaks (P1, P2) of the second order kernels in the nasal and temporal fields within the central 15° diameter.

### Results

The mean age of all participants (patients and normals) was 63.8 ± 10.8 years. With the progression of glaucoma, the amplitudes of P1 in the nasal hemifield increased and the amplitudes of P2 decreased. The nasal to temporal ratio (N/T ratio) of the P1 amplitudes and the negative slope of the line between P1 and P2 (P1P2 Slope) in the nasal field were larger at each glaucoma stage except at the PPG stage. Both the N/T amplitude ratio and P1P2 Slope were weakly but significantly correlated with the MD (r = -0.3139, P<0.0001; r = 0.4501, P<0.0001, respectively), and the OCT parameters (all P<0.0001) except the outer layer thickness.

### Conclusions

Our findings indicate that the amplitudes of P1 and P2 of the second order kernel of the mfERGs in the nasal field of the center region can be good markers for the stages of glaucoma.

**Competing interests:** The authors have declared that no competing interests exist.

## Introduction

Glaucomatous optic neuropathy is characterized by the death of the retinal ganglion cells (RGCs) and their axons. The method most used to assess and follow the progression of the damages in glaucomatous eyes is standard automated perimetry (SAP) with the Humphrey field analyzer (HFA). Standard flash electroretinograms (ERGs) have also been used to obtain objective assessments of the stage of glaucoma but the data collected did not provide critical information on the early glaucomatous changes. However, the pattern ERGs (pERGs) can be used as an adjunct to the diagnosis and management of glaucoma suspects, i.e., those with glaucomatous optic disc with normal fields [1–3]. The amplitudes of the photopic negative response (PhNR) which originates from the neural activities of the RGCs [4–7] have been shown to be reduced in glaucomatous eyes [2, 8–10].

Glaucomatous damage is characterized by a superior and inferior asymmetry, not nasal and temporal asymmetry, and the most frequently affected region of glaucomatous eyes at the early stage of disease is on the temporal retina [11]. In an earlier study, we analyzed the amplitude of the first positive peak (P1) of the second order kernel of the multifocal ERGs (mfERGs) by averaging relatively small areas with fully dilated pupils and contact lens electrodes [12]. Comparisons were made between the glaucomatous eyes with a superior or an inferior-dominant hemifield defects. Because the differences in the P1 amplitudes and the superior-inferior ratios of the superior or inferior hemispheres were not significant [12], we concluded that the amplitudes of the superior and inferior responses of the second order kernels did not reflect the visual field defects. However, when a comparison was made between the temporal and nasal hemifields within the central 5° radius, there was a statistically significant difference in the nasal-temporal (N/T) P1 amplitude ratio between normal eyes and glaucoma patients, and there was a significant increase of the P1 amplitude and a nasal-temporal asymmetry was lost in the glaucoma patients [12] as had been reported [13–17]. Thereafter, significant correlations were found between the ganglion cell complex (GCC) thickness, the mean sensitivities and the N/T P1 amplitude ratio, especially in the inferior and inferotemporal retinal areas (corresponding to superior and superonasal visual field) [18]. These findings indicated that the mfERGs might be helpful in determining the functional defects in glaucomatous eyes. However, there has not been a study published on the relationship between the amplitudes of the mfERGs and the stage of glaucoma. For further clinical application, it is important to ascertain whether mfERGs can be used to determine the stage of the glaucoma.

Thus, the purpose of this study was to determine whether the amplitudes of the P1 and P2 components of the second order kernel of the mfERGs were correlated with the different stages of glaucoma. To accomplish this, we recorded mfERGs with natural pupil using skin electrodes, and we focused on the nasal responses within the central 15° diameter.

## Materials and methods

This study was a prospective cross-sectional observational study conducted in the Glaucoma Service Clinic of the Saneikai Tsukazaki Hospital, in Himeji, Japan between May 21, 2019 to March 5, 2020. All procedures performed in these studies involving human participants were in accordance with the ethical standards of the Institutional and/or National Research Committee and with the 1964 Declaration of Helsinki and its later amendments or comparable ethical standards. A written informed consent was obtained from all participants for their information to be stored in the hospital. The medical records of all participants in the hosipial were also retrospectively reviewed. We selected 218 eyes of 132 patients with open angle glaucoma (OAG) and 62 eyes of 62 normal subjects. The experimental protocols were approved by

the Institutional Board of Research Associates of Gifu University Graduate School of Medicine and Saneikai Tsukazaki Hospital. Personal identifiers were removed from the records prior to data analysis. We reanalyzed the data obtained from all participants. Background data (age, sex, eye: right/left, intraocular pressure, refractive error, and medications) of all patients was also extracted from the medical record.

The individual pictured in Fig 1 has provided written informed consent (as outlined in PLOS consent form) to publish their image alongside the manuscript.

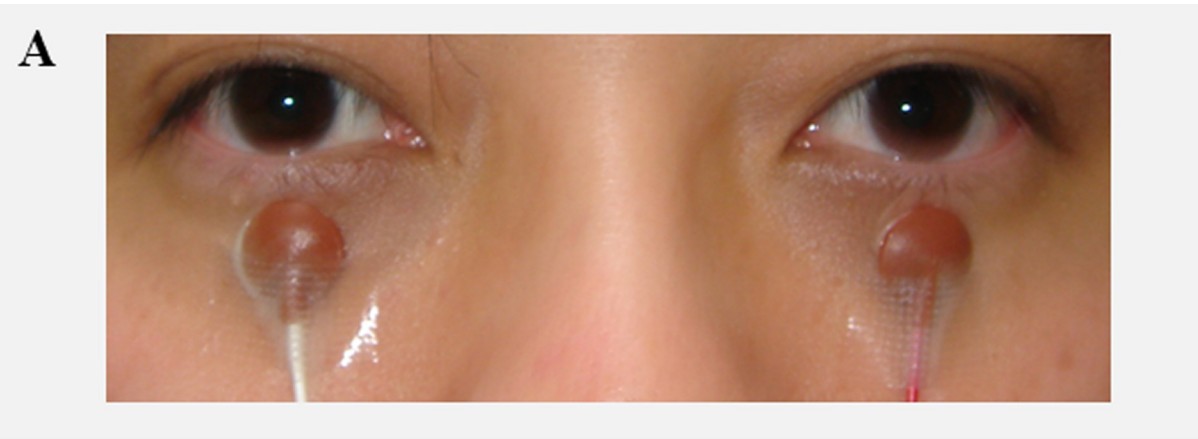

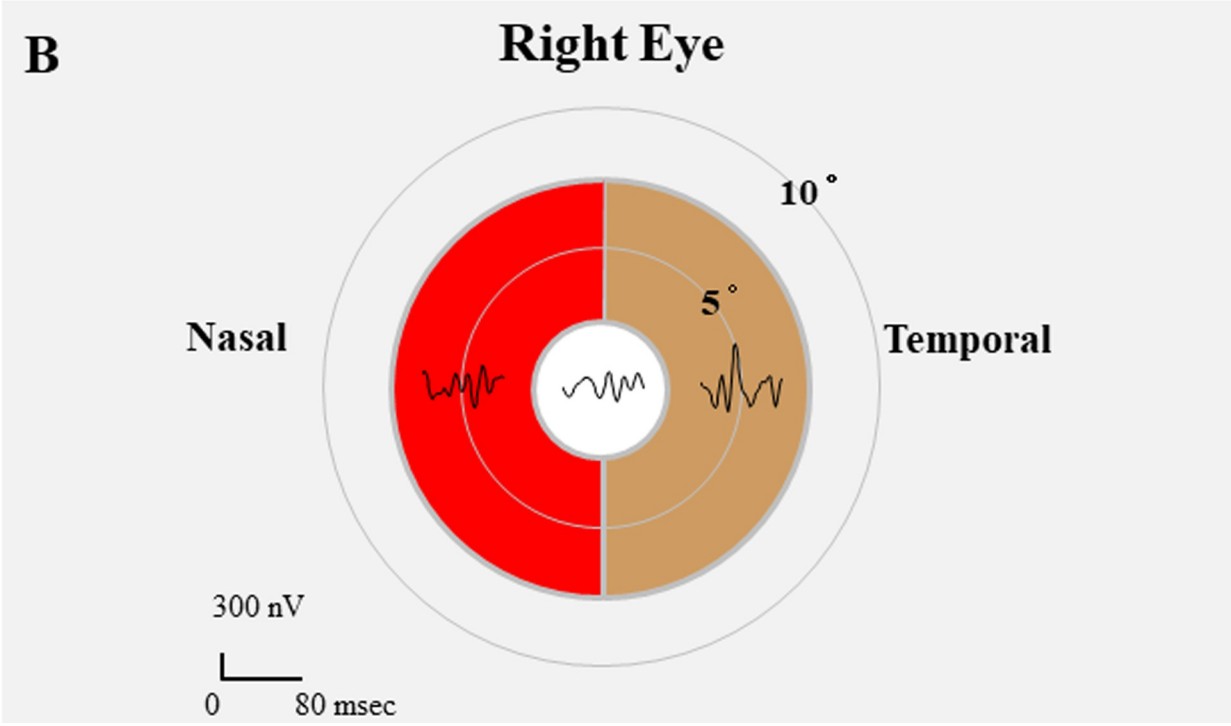

**Fig 1. Electrode positions used to record multifocal electroretinograms (mfERG).** A: The pupils were not dilated, and a skin electrode was placed on lower lid of each eye. The contralateral eye was not covered. A gold-cup electrode was placed on the right earlobe as the ground electrode. B: The mfERGs elicited by the 3 stimulus arrays within a circle of 7.5° radius. We separated the arrays according to the temporal (brown color) nasal (red color) hemisphere. For further analyses, the first slice of the second-order kernels was extracted from the mfERGs according to each stimulus array.

## Subjects

All participants were over 40-year-of-age and had a best-corrected visual acuity (BCVA) of 0.1 logarithm of the minimum angle of resolution (logMAR) units (20/25) or less. The refractive error (spherical equivalent) was measured with an autorefractometer (Topcon KR8900, Tokyo, Japan), and it ranged from -6.00 to +3.00 diopters (D). The exclusion criteria were: prior intraocular surgery, prior chorioretinal or vitreoretinal disease, and evidence of systemic disease such as diabetes mellitus or uncontrolled hypertension.

The diagnostic criteria for preperimetric glaucoma (PPG) and perimetric glaucoma were an AC angle greater than Grade 3 in the Shaffer's classification, the presence of characteristic glaucomatous changes in the optic disc with the retinal nerve fiber layer (RNFL) thinning in the OCT images, but without the presence of visual field defects detectable with the Humphrey Field Analyzer (HFA; 750 I Series, Carl Zeiss Meditec, Dublin, CA, USA) with the central 10–2 Swedish interactive threshold algorithm (SITA) program.

All of the patients with OAG except those with PPG had a normal open angle. In addition, all had visual field loss corresponding to the optic disc changes, and no other systemic neurologic abnormalities. Patients with normal tension glaucoma (NTG) and primary open-angle glaucoma (POAG) were included. In addition, the eligible subjects had to meet the following inclusion criteria: no ocular abnormalities except glaucoma, no history of any medication use that could affect the pupil diameter, and no prior ocular surgeries including laser therapy. The interval between the visual field examination, OCT imaging, and mfERG recordings was less than 3 months. The patients were classified based on the mean deviation (MD) values of the HFA 24–2 program according to the Anderson-Patella classification [19] into early stage glaucoma with MD $>$-6 decibels (dB), moderate stage glaucoma with $-6 \geq MD > -12$ dB, and advance stage glaucoma with $MD \leq$ -12 dB. If the patients who had not undergone HFA24-2 but took the HFA10-2 program were diagnosed as PPG or early glaucoma, the MD value of 10–2 program was used for classifying the glaucoma severity (The MD values were not used for statistical analysis): PPG (undetectable) and early stage glaucoma with MD $>$-6 dB.

The inclusion criteria for normal subjects were IOP $\leq$21 mmHg and normal ophthalmoscopic appearance of the optic disc. The normal subjects did not undergo HFA and optical coherence tomographic (OCT) testing.

## Optical coherence tomography (OCT)

A Cirrus High-Definition-OCT (HD-OCT) 5000 instrument was used to obtain the OCT images. The pupils were dilated to 8 mm with topical 0.5% tropicamide and 0.5% phenylephrine (Mydrin- P®: Santen Pharmaceutical, Osaka, Japan). The Ganglion Cell Analysis (GCA) and Macula Cube 200 x 200 programs were used. The peripapillary retinal nerve fiber layer (RNFL) thickness was measured automatically by the software for a diameter of 3.46 mm consisting of 256 A-scans centered on the optic disc. The average RNFL thickness of the circumference of the optic disc was used for the statistical analyses.

The macular cube scan generated one set of 200 horizontal B-scans, each comprised of 200 A-scans centered on a 6 x 6 mm macular region. The built-in GCA algorithm (Cirrus HD-OCT software, version 6.0) detected and measured the thicknesses of the macular ganglion cell-inner plexiform layer (GCIPL) and the outer layer (OL) within a 6 x 6 x 2 mm cube in an elliptical annulus around the fovea. The GCA algorithm identified the outer boundary of the RNFL and the inner plexiform layer (IPL). The OCT images used for the analyses had a signal strength $>$7/10.

The GCIPL thickness was measured as the distance from the outer border of the RNFL to the outer border of the IPL, and the OL thickness was measured as the distance between the outer border of the IPL and the outer border of the retinal pigment epithelium.

## Visual field testing

All of the patients with OAG had a perimetric examination with the Humphrey Field Analyzer with the Central 24–2 and the Central 10–2 SITA programs. All subjects had been tested earlier with the SITA standard testing procedures. Defects of the visual field resulting from OAG except the PPG were defined as those with glaucomatous hemifield test results outside the normal limits or pattern standard deviation of less than 5% probability of being normal on 2 consecutive SAP tests. Data from the examinations that met the reliability criteria of false-negative responses <15%, false-positive responses <15%, and fixation loss <20% were used in the statistical analyses.

## Multifocal ERGs (mfERGs, Fig 1)

The mfERGs were elicited with the LE-4100 mfERG stimulator (Mayo Corporation, Inazawa, Aichi, JAPAN), and the recordings were performed according to a published method [12, 13, 20, 21]. The pupils were not dilated, and a skin electrode was placed on the lower eyelid of both eyes (Fig 1A). The contralateral eye was not covered during the recording of one eye. A gold-cup electrode was attached to the right earlobe as the ground electrode. The refractive error of all subjects was corrected for their best-corrected visual acuity (BCVA) for a stimulus viewing distance of 16 cm. During the mfERG recordings, the subjects sat with their chin and forehead tightly fixed to the head holder frame. The subject was instructed to fixate a cross target at the center of the stimulus screen with the eye being stimulated. The amplitudes of the mfERGs are expressed as the response density, $nV/deg^2$, which represent the amplitudes as a function of the stimulus area.

## Nasal and temporal amplitudes of first slice of second-order kernel of mfERGs

The visual stimuli consisted of a central circle and nasal and temporal semi-annuli that were displayed on a mobile DLP projector (M115HD: Dell Inc., Round Rock, Texas, USA). The stimulus array subtended a visual angle of 15˚ diameter. The radius of the central circle was 2.5˚ (Fig 1B). The circle and semi annuli of the stimulus were independently alternated between black (5 $cd/m^2$) and white (1,500 $cd/m^2$; contrast, 95.1%) at a frame rate of 75 Hz according to a binary m-sequence. The band pass filters were set at 10 to 100 Hz. The position of the eye during the recordings was monitored through the recording window. Each recording lasted approximately 2 mins which consist of four 30 seconds, and periods with eye movements or blinks artifacts were recorded again. An artifact elimination technique and spatial smoothing were not used. A signal-to-noise ratio (SNR) ≥0 dB was accepted for the mfERG measurements. The amplitudes of the first positive peak, P1 and the second positive peak, P2 were studied (Fig 2A). The amplitudes of P1 and P2 of the first slice of the second order kernel was measured according to a published method [13, 18].

## Analyses of amplitudes of mfERGs

The mfERGs elicited by the 3 stimulus arrays within a circle of 7.5˚ radius are shown in Fig 1B. We analyzed the first slice of the second-order kernel of mfERGs. The amplitudes of P1 or P2 from the nasal hemisphere were compared with the corresponding amplitudes in the temporal

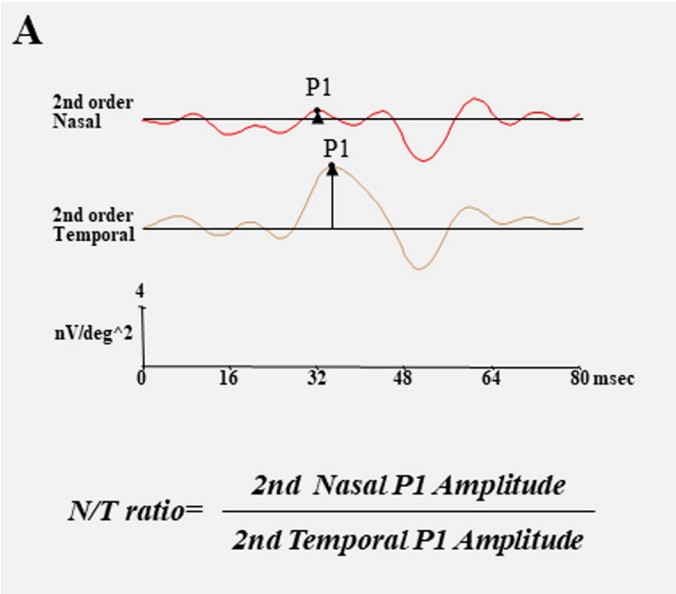 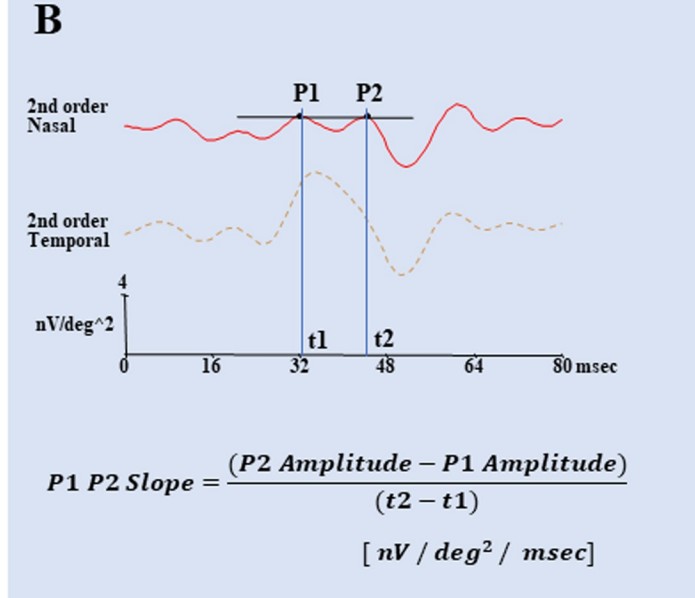

**Fig 2. Ratio of the P1 amplitudes of the mfERGs of the nasal to the temporal hemisphere (N/T amplitude ratio) within the central 7.5° radius.** A: Ratio is used to evaluate the asymmetry between the nasal and temporal fields. B: Slope of the line between P1 and P2 (P1P2 Slope) of the second-order kernel in the nasal field was used to evaluate the degree of asymmetry.

hemisphere. We also calculated the ratio of the amplitudes of P1 between the nasal to temporal (N/T) hemisphere [11] (Fig 2A), and the slope of the line between P1 and P2 (P1P2 Slope) in the nasal field (Fig 2B).

## Statistical analyses

When both eyes of a normal subject met the inclusion criteria, one eye was randomly selected for the statistical analyses. The relationships between the parameters were assessed by the coefficients of correlation and linear regression analyses. The Mann-Whitney U and Chi-squared tests were used to determine the significance of the differences of the values between normal participants and glaucoma patients. Spearman rank correlation coefficients were used to assess the relationships among the visual field data, OCT data, N/T hemisphere P1 amplitude ratio, and P1P2 Slope of the nasal responses. A $P$ value of <0.05 was taken to be significant. All statistical analyses were performed using SPSS software version 16.0 (SPSS Japan, Tokyo, Japan.).

## Results

The demographics of the OAG patients and normal subjects are presented in Table 1. The values are the means ± standard deviations (SDs).

### Amplitudes of P1 and P2 of first slice of second-order kernel of mfERGs in nasal hemifield (Tables 2 and 3 and Fig 3A and 3B)

The mean amplitude of P1 (in nV/deg$^2$) for all OAG eyes was 1.912 ± 1.117, for PPG was 1.486 ± 0.945, early stage was 1.744 ± 1.067, moderate stage was 2.199 ± 1.077, and advanced stage was 2.386 ± 1.223. The amplitude of P1 of the normal subjects was 1.203 ± 0.917 nV/deg$^2$. The mean latency of P1 of all eyes with OAG was 32.72 ± 2.73 msec, and that for PPG eyes was 32.42 ± 3.61 msec, early stage was 32.64 ± 2.64 msec, moderate stage was 32.93 ± 2.67

**Table 1. Demographic data of all participants.**

| Variable | OAG | | | | | normal |
|---|---|---|---|---|---|---|
| | **Total** | **PPG** | **early** | **moderate** | **advanced** | |
| Subjects (case/eye) | 132/218 | 19/20 | 91/121 | 38/41 | 30/36 | 62/62 |
| Sex:eye (male/female) | 62 /70 | 15/4 | 37/54 | 18/20 | 14/16 | 26/36 |
| Eye (Right/Left) | 102/116 | 6/14 | 56/65 | 21/20 | 19/17 | 46/16 |
| Age [years] | 64.7 ±10.6 (41–86) (P = 0.0080) | 58.2±10.8 (43–83) (P = 0.3755) | 63.7±9.6 (41–83) (P = 0.0559) | 68.0±10.7 (41–83) (P = 0.0007) | 67.9±11.5 (41–86) (P = 0.0020) | 60.5± 10.7 (44–82) (P value; vs normal) |
| RE [Diopters] | -1.772± 2.117 (-6.00 -+2.38) | -1.344± 2.007 (-4.88 -+2.13) | -1.854± 2.229 (-6.00 -+1.75) | -1.768± 1.886 (-6.00 -+1.63) | -1.605± 2.024 (-5.63 -+2.38) | ND |
| Intraocular Pressure [mmHg] | 14.7± 2.5 (8–24) | 16.0±3.1 (12–22) | 14.7±2.7 (7–24) | 14.5±2.4 (10–19) | 13.9±1.8 (10–18) | ND |
| Medications (Eyes With/Without) | 186/32 | 6/14 | 110/11 | 39/2 | 31/5 | - |
| HFA Central 24–2 Program MD [dB] | -5.790 ± 6.055 (-27.18 - +3.19) | -0.518 ± 1.506 (-3.49 - +1.68) | -2.219 ± 1.950 (-5.85 - +3.19) | -8.532 ± 1.567 (-11.83 - -6.05) | -16.620 ± 4.224 (-27.18 - -12.03) | ND |
| CpRNFL thickness [μm | 71.0± 10.7 (46–108) | 80.3±11.1 (46–100) | 72.6±9.9 (48–108) | 69.4±10.1 (53–108) | 62.4±7.5 (46–79) | ND |
| GCIPL thickness [μm]/ | 69.2 ± 8.5 (48–106) | 71.9±8.4 (51–86) | 71.4±8.2 (51–106) | 65.8±7.7 (48–80) | 64.1±7.5 (49–77) | ND |
| mRNFL thickness [μm]/ | 27.9 ± 5.3 (12–51) | 30.6±3.3 (26–36) | 28.8±4.4 (17–40) | 28.0±6.3 (14–51) | 23.2±5.0 (12–38) | ND |
| OL thickness [μm] | 128.1 ± 10.4 (83–173) | 130.6±11.9 (104–159) | 127.4±8.1 (105–151) | 129.2±12.0 (107–173) | 127.9±13.8 (83–158) | ND |

OAG: Open angle glaucoma

PPG: Preperimetric and perimetric glaucoma

RE: Refractive error, (spherical equivalent)

ND: no data

HFA: Humphrey Field Analyzer

MD: Mean Deviation

cpRNFL: Circumpapillary retinal nerve fiber layer thickness

GCIPL: Ganglion cell-inner plexiform layer

mRNFL: Macular retinal nerve fiber layer

Values are the means ± standard deviations (range)

Mann-Whitney U test

Among 6 eyes without the HFA24-2 program, 3 eyes were categorized in PPG and 3 eyes were categorized in early glaucoma using the HFA10-2 program.

msec, and advanced stage was 32.94 ± 2.60 msec. The latency of P1 of the normal subjects was 32.63 ± 3.53 msec.

The amplitude of P1 in the nasal hemifield was significantly larger at each stage of glaucoma than in the normal group except in the PPG group ($P = 0.0012$, $P < 0.0001$, and $P < 0.0001$, respectively, Table 2). In addition, the P1 amplitude was significantly larger in the Moderate and Advanced groups than in the PPG group ($P = 0.0340$ and $P = 0.0183$, Table 3). It was also significantly larger in the Moderate and Advanced groups than in the Early group ($P = 0.0466$ and $P = 0.0109$, Table 3).

The amplitudes of P2 (in nV/deg$^2$) in all eyes with OAG was 0.479 ± 1.197, PPG was 1.174 ± 1.261, early stage was 0.556 ± 1.167, moderate stage was 0.158 ± 1.110, and advanced stage was 0.201 ± 1.201. The amplitude of P2 in normal subjects was 1.017 ± 1.186 nV/deg$^2$. The latency of P2 in msec of all eyes with OAG was 44.87 ± 2.62, PPG was 45.83 ± 3.66, early stage was 44.63 ± 2.47, moderate stage was 45.18 ± 1.88, and advanced stage was 44.81 ± 3.05 msec. The latency of P2 of the normal subjects was 44.97 ± 2.81 msec.

**Table 2. Amplitudes of P1 and P2 of the mfERGs in the nasal and temporal hemifields.** The P1 N/T amplitude ratio of the second-order kernel, P1P2 Slope of the nasal response of the second-order kernel.

| Variable | Nasal hemifield | | | | Temporal hemifield | | | | N/T ratio (*P* value; vs normal) | P1P2 Slope (*P* value; vs normal) |
|---|---|---|---|---|---|---|---|---|---|---|
| | P1 (*P* value; vs normal) | | P2 (*P* value; vs normal) | | P1 (*P* value; vs normal) | | P2 (*P* value; vs normal) | | | |
| | Amplitude (nV/deg$^2$) | Latency (msec) | Amplitude (nV/deg$^2$) | Latency (msec) | Amplitude (nV/deg$^2$) | Latency (msec) | Amplitude (nV/deg2) | Latency (msec) | | |
| PPG | 1.486 ± 0.945 (*P* = 0.3531) | 32.42 ± 3.61 (*P* = 0.8751) | 1.174 ± 1.261 (*P* = 0.7542) | 45.83 ± 3.66 (*P* = 0.4215) | 3.023 ± 1.107 (*P* = 0.4180) | 35.50 ± 3.26 (*P* = 0.0812) | 0.344± 0.728 (*P* = 0.0560) | 44.63 ± 2.53 (*P* = 0.1527) | 0.544± 0.375 (*P* = 0.5382) | -0.016 ± 0.116 (*P* = 0.7624) |
| Early | 1.744 ± 1.067 (*P* = 0.0012) | 32.64 ± 2.64 (*P* = 0.9151) | 0.556 ± 1.167 (*P* = 0.0074) | 44.63 ± 2.47 (*P* = 0.8714) | 2.464 ± 1.087 (*P* = 0.2017) | 34.26 ± 2.82 (*P* = 0.5946) | 0.581 ± 1.317 (*P* = 0.1167) | 44.00 ± 2.77 (*P* = 0.3085) | 0.740± 0.537 (*P* = 0.0001) | -0.115± 0.154 (*P*<0.0001) |
| Moderate | 2.199 ± 1.077 (*P*<0.0001) | 32.93 ± 2.67 (*P* = 0.5697) | 0.158 ± 1.110 (*P* = 0.0006) | 45.18 ± 1.88 (*P* = 0.3613) | 2.610 ± 1.252 (*P* = 0.7413) | 34.82 ± 2.80 (*P* = 0.4500) | 0.270 ± 1.189 (*P* = 0.0079) | 44.53 ± 2.68 (*P* = 0.1425) | 0.861± 0.664 (*P* = 0.0001) | -0.181± 0.174 (*P*<0.0001) |
| Advanced | 2.386 ± 1.223 (*P*<0.0001) | 32.94 ± 2.60 (*P* = 0.6465) | 0.201 ± 1.201 (*P* = 0.0046) | 44.81 ± 3.05 (*P* = 0.9057) | 2.436 ± 1.146 (*P* = 0.2240) | 33.61 ± 2.86 (*P* = 0.5189) | 0.489 ± 1.498 (*P* = 0.0873) | 44.88 ± 2.65 (*P* = 0.0189) | 1.077± 0.679 (*P*< 0.0001) | -0.210± 0.155 (*P*< 0.0001) |
| Total OAG | 1.912 ±1.117 (*P*<0.0001) | 32.72 ± 2.73 (*P* = 0.0765) | 0.479 ± 1.197 (*P* = 0.0016) | 44.87 ± 2.62 (*P* = 0.7610) | 2.538 ± 1.135 (*P* = 0.3093) | 34.37 ± 2.89 (*P* = 0.5149) | 0.486 ± 1.282 (*P* = 0.0177) | 44.30 ± 2.72 (*P* = 0.0724) | 0.800± 0.590 (*P*< 0.0001) | -0.134± 0.163 (*P*< 0.0001) |
| Normal | 1.203 ± 0.917 | 32.63 ± 3.53 | 1.017± 1.186 | 44.97 ± 2.81 | 2.785 ± 1.350 | 33.95 ± 3.43 | 0.985 ± 1.310 | 43.65 ± 2.93 | 0.504± 0.463 | -0.011 ± 0.147 |

OAG: Open angle glaucoma

N/T: nasal to temporal amplitude ratio

PQPW Slope: slope of between P1 and P2

Values are mean ± standard deviation

Mann-Whitney U test

The amplitude of P2 in the nasal hemifield was significantly smaller in all glaucoma groups than in the normal group except in the PPG group (*P* = 0.0074, *P* = 0.0006, and *P* = 0.0046, respectively, Table 2). In addition, the P2 amplitude was significantly smaller in the Early, Moderate, and Advanced groups than in the PPG group (*P* = 0.0491, *P* = 0.0054, and *P* = 0.0167, respectively, Table 3).

**Table 3. Comparisons of mfERG parameters between the groups.**

| | | | PPG vs Early | PPG vs Moderate | PPG vs Advanced | Early vs Moderate | Early vs Advanced | Moderate vs Advanced |
|---|---|---|---|---|---|---|---|---|
| Nasal hemifield | P1 | Amplitude | 0.3722 | 0.0340 | 0.0183 | 0.0466 | 0.0109 | 0.4685 |
| | | Latency | 0.7438 | 0.5364 | 0.6426 | 0.5250 | 0.7353 | 0.7382 |
| | P2 | Amplitude | 0.0491 | 0.0054 | 0.0167 | 0.0793 | 0.2058 | 0.8382 |
| | | Latency | 0.2819 | 0.9195 | 0.5474 | 0.1505 | 0.7190 | 0.6510 |
| Temporal hemifield | P1 | Amplitude | 0.0526 | 0.3255 | 0.0699 | 0.3829 | 0.7829 | 0.4623 |
| | | Latency | 0.0916 | 0.2971 | 0.0345 | 0.6743 | 0.2279 | 0.2481 |
| | P2 | Amplitude | 0.3948 | 0.5593 | 0.9319 | 0.0868 | 0.4752 | 0.6830 |
| | | Latency | 0.3239 | 0.9508 | 0.8089 | 0.2969 | 0.0470 | 0.5756 |
| N/T ratio | | | 0.0594 | 0.0221 | 0.0010 | 0.3384 | 0.0066 | 0.0616 |
| P1P2 Slope | | | 0.0018 | 0.0001 | <0.0001 | 0.0274 | 0.0018 | 0.8340 |

mfERG, multifocal electroretinogram;

PPG, preperimetric and perimetric glaucoma;

N/T, nasal to temporal amplitude ratio;

P1P2 Slope, slope of line between P1 and P2

Mann–Whitney U test

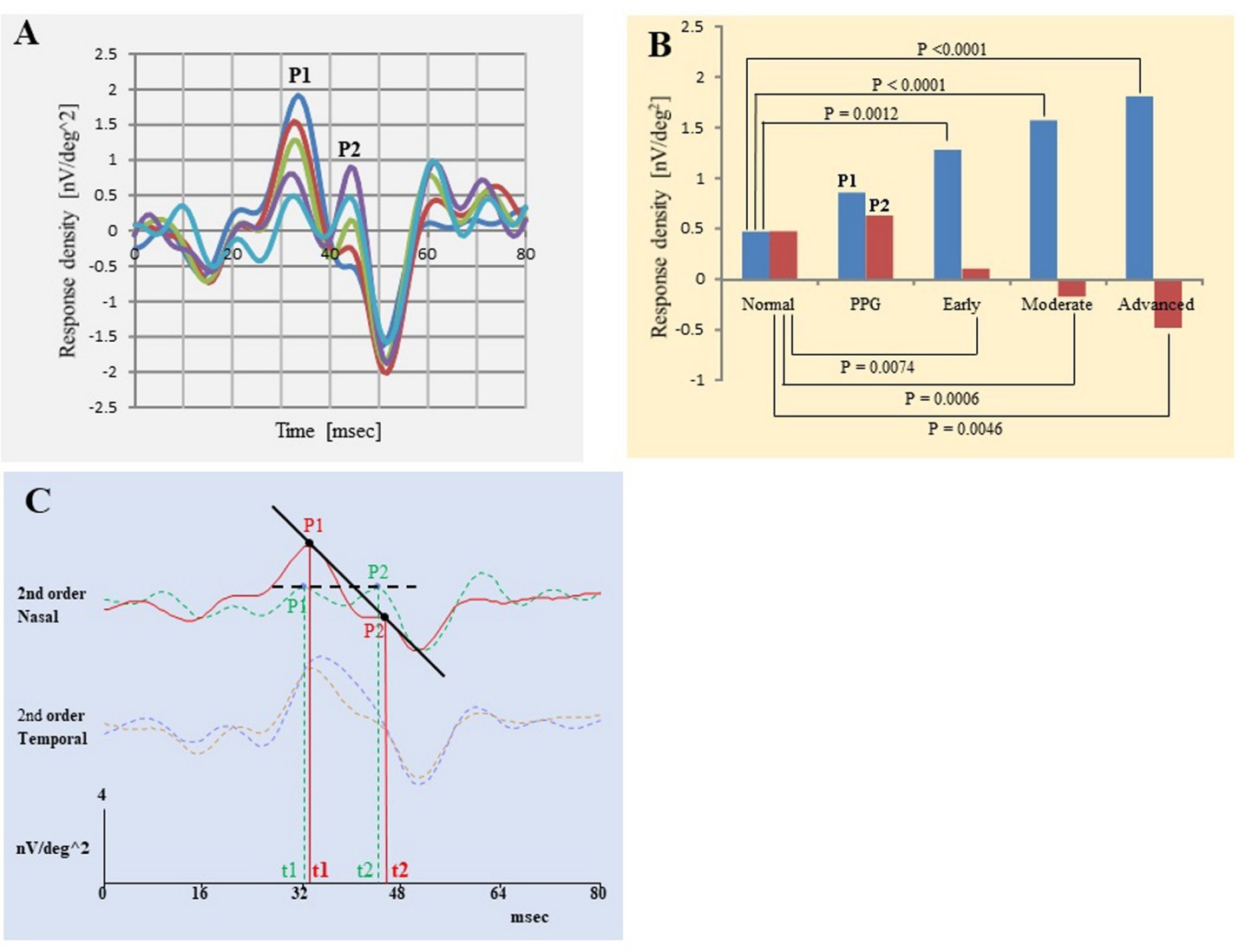

**Fig 3. Averaged first slice of second-order kernel of the mfERGs of the nasal hemisphere within the central 7.5° for different stages of glaucoma (Blue trace: Advanced stage, Brown trace: Moderate stage, Light green trace: Early stage, Purple trace: PPG stage, and Light blue trace: Normal).** A and B: Amplitude of P1 (B: Blue bar) increased, but amplitude of P2 (B: Brown bar) decreased with the progression of glaucoma stage. C: Slope of the line between P1 and P2 (P1P2 Slope) of the nasal response of the second-order kernel (Red line: the nasal response of the second-order kernel of one advanced case. Green broken line is response of one normal subject. The P1P2 Slope on the advanced glaucoma is negatively increased compare with normal subject.

### Nasal to temporal (N/T) ratio of P1 amplitude (Tables 2 and 3)

The N/T ratio of the P1 amplitude of all OAG eyes was 0.800 ± 0.590, PPG was 0.544 ± 0.375, early was 0.740 ± 0.537, moderate was 0.861 ± 0.664, and advanced was 1.077 ± 0.679. The N/T ratio for the normal subjects was 0.504 ± 0.463. The N/T ratio of P1 was significantly larger in each glaucoma group except PPG than in normal ($P$ = 0.0001, $P<$0.0001, and $P<$0.0001, respectively, Table 2). In addition, The N/T ratio was significantly larger in the Moderate and Advanced group than in the PPG group ($P$ = 0.0221 and $P$ = 0.0010, Table 3), and it was significantly larger in the Advanced group than in the Early group ($P$ = 0.0066, Table 3). No significant correlation was found between age and N/T ratio in normal subjects (r = 0.0392, $P$ = 0.7624; Spearman rank correlation, not shown in the Tables).

## Slope of line between P1 and P2 (P1P2 Slope) of nasal responses (Tables 2 and 3 and Fig 3C)

The P1P2 Slope of all eyes with OAG was -0.134 ± 0.163, in eyes with PPG it was -0.016 ± 0.116, early stage was -0.115 ± 0.154, moderate stage was -0.181 ± 0.174, and advanced stage was -0.210 ± 0.155. The slope in normal subjects was -0.011 ± 0.147. The negative P1P2 Slope was significantly larger in each glaucoma group than in normal group except in the PPG group (all $P < 0.0001$, Table 2). In addition, the negative P1P2 Slope was significantly larger in the Early, Moderate, and Advanced groups than in the PPG group ($P = 0.0018$, $P = 0.0001$, and $P < 0.0001$, respectively, Table 3), and it was significantly larger in the Moderate and Advanced groups than in Early group ($P = 0.0274$ and $P = 0.0018$, Table 3). No significant correlation was found between the age and P1P2 Slope in normal subjects ($r = 0.1509$, $P = 0.2417$; Spearman rank correlation, not shown in the Tables).

## Correlations between MD of humphrey central 24–2 program and N/T of P1 amplitude ratios and P1P2 slope of nasal responses of second-order Kernel in eyes with OAG (Table 4)

The N/T amplitude ratio was significantly and negatively correlated with the MD of HFA24-2 ($r = - 0.3139$, $P < 0.0001$; Spearman rank correlation).

The P1P2 Slope was significantly correlated with the MD of HFA24-2 ($r = 0.4501$, $P < 0.0001$).

## Correlation between OCT parameters and N/T of P1 Amplitude Ratios and P1P2 slope of nasal responses of second-order Kernel in eyes with OAG (Table 4)

The N/T amplitude ratio was significantly correlated with the average thickness of the GCIPL in the macular area ($r = -0.2798$, $P < 0.0001$). The N/T ratio was significantly correlated with

**Table 4. Correlation between N/T amplitude ratio of second order kernel and the slope of the line and MD of Humphrey Central 24–2 program. OCT findings are also presented.**

| Variable | HFA 24–2 | | Macular thickness | | | | | | | |
| | MD | | GCIPL | | mRNFL | | OL | | cpRNFL | |
| | r | P | r | P | r | P | r | P | r | P |
| Nasal P1 | -0.3358 | <0.0001 | -0.2456 | <0.0001 | -0.3476 | <0.0001 | -0.01260 | 0.8345 | -0.2755 | <0.0001 |
| Nasal P2 | 0.2562 | <0.0001 | 0.1398 | 0.0204 | 0.2256 | 0.0002 | -0.08250 | 0.1713 | 0.1568 | 0.0094 |
| N/T ratio | -0.3139 | <0.0001 | -0.2798 | <0.0001 | -0.4110 | <0.0001 | 0.0459 | 0.4464 | -0.3230 | <0.0001 |
| P1P2 Slope | 0.4501 | <0.0001 | 0.2803 | <0.0001 | 0.3604 | <0.0001 | -0.0345 | 0.5671 | 0.3277 | <0.0001 |

N/T: nasal to temporal amplitude

P1P2 Slope: slope of line between P1 and P2

OCT: optical coherence tomography

HFA: Humphrey Field Analyzer

MD: Mean Deviation

GCIPL: Ganglion cell-inner plexiform layer

mRNFL: Macular retinal nerve fiber layer

OL: Outer Layer

CpRNFL: Circumpapillary retinal nerve fiber layer thickness

Spearman's rank correlation

the average mRNFL thickness (r = -0.4110, P <0.0001). The N/T amplitude ratio was significantly correlated with the average thickness of the cpRNFL (r = - 0.3230, $P$ <0.0001). The correlation between the N/T amplitude ratio and the average thickness of the OL was not significant (r = 0.0459, $P$ = 0.4464).

The P1P2 Slope was significantly correlated with the average macula thickness of the GCIPL (r = 0.2803, $P$ <0.0001). The P1P2 Slope was significantly correlated with the average mRNFL (r = 0.3604, $P$ <0.0001). The P1P2 Slope was significantly correlated with the average thickness of the cpRNFL (r = 0.3277, $P$ <0.0001). The correlation between the P1P2 Slope and average thickness of the OL was not significant (r = -0.0345, $P$ = 0.5671).

## Discussion

Our results showed that the amplitude of P1 of the first slice of the second-order kernel of the mfERGs in the nasal hemifield was significantly larger in more advanced stages of glaucoma (Table 2 and Fig 3A and 3B). On the other hand, the amplitude of the P2 was significantly smaller in more advanced stages of glaucoma (Table 2 and Fig 3A and 3B). The N/T ratio of the P1 amplitude was significantly larger (Tables 2 and 3), and the negative slope of the line was significantly steeper in more advanced stages of glaucoma (Tables 2 and 3 and Fig 3C). There was a weak but significantly difference in each N/T amplitude ratio or P1P2 Slope between the normal subjects and the glaucoma groups except for the PPG group (Tables 2 and 3). Both the N/T amplitude ratio and P1P2 Slope were also weakly correlated with the MD, and the OCT parameters except the OL thickness (Table 4). The amplitudes of P1 and P2 in the nasal visual field of the second order response of the mfERG were a good marker of the stage of glaucoma.

There have been several recent studies that examined the paracentral region of the retina that has more glaucomatous damage in the early stages of OAG [14, 22–25]. The nasal-temporal asymmetry of the mfERGs is known to be altered in glaucomatous eyes [13–17]. Earlier, we recorded mfERGs with a dilated pupil using contact lens electrodes, and our findings showed significant differences in the N/T amplitude ratio of the first slice of the second-order kernel of the mfERGs with stimulus hexagons on the central 5˚ radius between normal subjects and NTG patients. In addition, the N/T amplitude ratio of the central 5˚ radius and the antilog averaged value of each parameter obtained with HFA central 30–2 were significantly correlated [12]. Moreover, the N/T amplitude ratio was significantly correlated to the unique glaucomatous visual field defects in which eyes at the early stages of OAG had defects in the superior-central and superior-nasal visual fields [20, 26, 27]. We found that the results at the N/T hemisphere amplitude ratio within a circle of 7.5˚ radius were similar to the N/T amplitude ratio of the hexagons in the 5˚ radius.

The P1 and P2 components of the first slice of the second order kernel were elicited by a circular stimulus of the 7.5˚ region in OAG patients especially in the nasal hemifield. The results showed that the P1 amplitude was larger but the P2 amplitude was smaller in the OAG patients (Fig 3A and 3B). The differences in the amplitudes of the P1 component in the nasal hemifield between normal and OAG subjects may be because the second-order kernel response of the mfERGs results from responses from the retinal components (RCs) and the optic nerve head components (ONHCs) [15]. These two components with opposite polarities are canceled at the central region in normal subjects. The P1 components become more distinct due to the decrease or loss of the ONHCs and the survival of the RCs in OAG patients. Because the ganglion cell/photoreceptor ratio is larger at the center [15, 28], we suggest that the P1 component would be a helpful objective test to assess the visual function in OAG patients. However, the origin of the P2 component has not been determined, and it is assumed that there is an

association of the P1 with P2 components. In the advanced OAG patients, the area under the ROC curve of the N/T amplitude ratio was 0.786 and the P1P2 Slope was 0.832 (not shown in this paper). These results indicate that the sensitivity of the P1P2 Slope is slightly better than that of N/T amplitude ratio to assess the stage of the glaucoma.

We recorded the mfERGs with natural pupils, the black luminance was 5 cd/m$^2$, and the white luminance was 1,500 cd/m$^2$ for the multifocal stimuli. The mfERG guidelines recommended a stimulus luminance of 100–200 cd/m$^2$ to elicit and record the mfERGs with fully dilated pupils [21]. Our recordings were made from non-dilated eyes because eyes with glaucoma can have incomplete pupil dilatation because of posterior synechiae after glaucoma surgery or senile miosis. In addition, mydriasis is contraindicated in eyes with narrow angle eye and in patients with plateau iris. The recording with natural pupil may solve some of those problems, but the retinal illumination is reduced which affects the amplitude and prolongs the peak latency [29]. Poloschek reported that the effective retinal illuminance (Troland) is decreased by a factor of 5.4 when recording the mfERGs with natural pupils of 3.7 mm diameter [30]. The retinal illuminance depends on the pupillary area and the photopic or scotopic luminance [31]. In this study, the pupils during the elicitation of the mfERG with mydriasis and non- mydriasis were photographed with an infrared camera to measure the pupillary diameter which was 7.5 mm and 2 mm, respectively. Using the Moon et al. formula, the effective pupillary area was 23.6 mm$^2$ and 3.01 mm$^2$, respectively [32]. The ratio of the effective pupil area in mydriasis to non-mydriasis was 7.84. Therefore, the luminance at non-mydriasis which is equivalent to 200 cd/m$^2$ at mydriasis, is 200 x 7.84 = 1,568 cd/m$^2$ and 1,500 cd/m$^2$ was used in this study.

We also recorded mfERGs using skin electrodes. Skin electrodes are not generally recommended as the active recording electrodes because of the reduction in the amplitude of the responses and the high variability of the results in physiologically normal subjects [33]. Skin electrodes are known to produce lower amplitude responses than that of conventional contact lens electrodes [33]. On the other hand, the placement of the corneal electrodes requires a skilled technician and is uncomfortable for some patients [34]. In addition, corneal electrodes have a potential of causing corneal abrasions, and children tend not to cooperate during the insertion of the corneal lens electrode [33, 35]. It has been shown that skin electrodes with signal averaging can provide satisfactory results with high quality ERGs [35, 36]. Earlier studies have also shown that skin electrodes have recording repeatability comparable to the contact lens electrodes [34]. Because we performed the mfERG recordings with skin electrodes and without anesthesia, the ERGs recordings can be performed in children [34, 37]. Another advantage of the skin electrodes is the effects of the anesthetic agents on the ERGs [38] can be avoided. Moreover, it can be used for glaucoma patients with a filtering bleb. These findings suggest that skin electrodes is a good option for recording the mfERGs.

There are some limitations of our study. First, the number of OAG patients in each group except in the early glaucoma group was small. Second, we did not take into account the effect of sex and age on the glaucomatous changes. The patients with moderate and advanced glaucoma were significantly older than the normal subjects (Table 1). Nabeshima reported that the response densities of the second order kernel significantly decreased and its peak latency was significantly prolonged in subjects above 50-years of age compared to 10- to 40-year-olds [39]. Then, the amplitude of each P1 or P2 may be influenced by age. In this study, since the average age of the subjects was around 60 years, and the methods of comparing amplitudes used ratio and slope, the effect of age was considered to be small. Third, we set the SNR to 0 dB as the criterion of the acceptable recording data. This 0 dB means that the root mean square (RMS) of the signal and one of the noise are equal. As a result, there was a lot of variations in the waveforms due to including completely different ones from the averaged waveform. To reduce the

variations, the SNR of 3 dB can be set as the criterion. If the SNR is less than 3 dB, the mfERG recordings (2-minute) will be performed twice, and these responses will be averaged to improve the SNR. The current recording condition is a monocular recording with a reference electrode to the opposite eye. If the binocular recording can be realized with a new reference electrode position, the total recording time will not be changed with both eyes of 4 minutes. Because the amplitudes of P1 and P2 on the waveform are measured manually, it is occasionally difficult to determine where P1 and P2 exist in the noisy waveform. A deep learning algorithm will solve this issue for the automatic measurements to extract the characteristic property of the second order kernel.

In conclusion, the amplitudes of the P1 and P2 of the first slice of the second order kernel in the nasal field of the center region of the mfERGs which were recorded with natural pupils and skin electrodes are significantly correlated with the stage of glaucoma. These findings indicate that the amplitudes of the P1 and P2 can be markers for the functional status of the inner retinal layer and the RGCs. In the future, the ERG technology including N/T amplitude ratio and slope of line of the mfERGs may prove to be useful and be a sensitive method for an objective determination of the function to assess and detect changes in glaucomatous eyes.

## Supporting information

**S1 Data.**
(XLSX)

**S2 Data.**
(XLSX)

## Acknowledgments

We would like to thank all of the participants involved in this study. We thank Professor Emeritus Duco Hamasaki of the Bascom Palmer Eye Institute for editing this manuscript. The authors thank Eiichiro Nagasaka for technical help in electrophysiological evaluations. We thank Hiroki Tanaka, MD, Mao Tanabe, PhD, MD, and Kyoko Ishida PhD, MD for help in data curation and formal analysis. We thank Etsuko Terao, Kanae Ueda, Yui Kimura, Satomi Oogi, Yuki Nagata, Yasuko Fujisawa, Saki Dote, Ryota Aoki, and Tomohiro Shojo for the ophthalmic examinations.

## Author Contributions

**Conceptualization:** Naoya Moroto, Kiyofumi Mochizuki.

**Data curation:** Shunsuke Nakakura, Hitoshi Tabuchi, Kiyofumi Mochizuki, Yusuke Manabe.

**Formal analysis:** Shunsuke Nakakura, Hitoshi Tabuchi, Kiyofumi Mochizuki, Yusuke Manabe.

**Methodology:** Kiyofumi Mochizuki.

**Project administration:** Kiyofumi Mochizuki.

**Supervision:** Kiyofumi Mochizuki, Hirokazu Sakaguchi.

**Validation:** Shunsuke Nakakura, Hitoshi Tabuchi, Kiyofumi Mochizuki.

**Writing – original draft:** Naoya Moroto.

**Writing – review & editing:** Naoya Moroto.

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
