## [Decision Letter · Decision Letter 0]

23 May 2022

PONE-D-22-03506Use of multifocal electroretinograms to determine stage of glaucomaPLOS ONE

Dear Sir,

Thank you for submitting your manuscript to PLOS ONE. After careful consideration, we feel that it has merit but does not fully meet PLOS ONE’s publication criteria as it currently stands. Therefore, we invite you to submit a revised version of the manuscript that addresses the points raised during the review process.

The reason for including nasal temporal asymmetry has to be detailed more to express why this is more imp than sup inferior asymmetry. Further, grammar and typos need significant editing using a professional service

We look forward to receiving your revised manuscript.

Kind regards,

Aparna Rao

Academic Editor

PLOS ONE

Journal Requirements:

2. We note that you state that consent was obtained from patients in your retrospective study. Please clarify the nature of the informed consent in your Ethics Statement and Methods section. When did patients provide their consent? Did patients consent to the medical treatment, and/or did they specifically consent to participate in this study or to have their medical records used in research? Were data anonymized/de-identified before access by researchers or did the ethics committee waive the need for additional informed consent?

Reviewers' comments:

Reviewer's Responses to Questions

**Comments to the Author**

1. Is the manuscript technically sound, and do the data support the conclusions?

Reviewer #1: No

Reviewer #2: Partly

2. Has the statistical analysis been performed appropriately and rigorously? 

Reviewer #1: Yes

Reviewer #2: Yes

3. Have the authors made all data underlying the findings in their manuscript fully available?

Reviewer #1: Yes

Reviewer #2: Yes

4. Is the manuscript presented in an intelligible fashion and written in standard English?

Reviewer #1: No

Reviewer #2: Yes

5. Review Comments to the Author

Reviewer #1: The authors investigated the role of multifocal electroretinograms to determine stage of glaucoma. They used the amplitudes of positive peaks (P1, P2) of the second order kernels in the nasal and temporal fields within central 7.5° diameter, reporting that the parameters were significantly different among normal and glaucoma, and glaucoma groups classified by glaucoma severity.

Although their study is interesting, it is necessary to suggest the sufficient rationale for assessing nasal side of the macular lesion and nasal to temporal asymmetry in glaucoma.

Glaucomatous damage is characterized by the superior and inferior asymmetry, not nasal and temporal asymmetry, and most frequently affected lesion of glaucoma at the early stage of disease is the temporal side (nasal step defect in early glaucoma).

Reviewer #2: This is an interesting study evaluation of the " To determine whether multifocal electroretinograms (mfERGs) recorded with natural

pupils and skin electrodes can be used to assess the stage of open angle glaucoma

(OAG). " The scientific work done commendable. Overall manuscript is well-written and easy to read with good explanations and interesting findings.

1- There are so many grammatical and syntax errors that need to be addressed.

2- However, the pattern ERGs (pERGs) have been shown to managing glaucoma.” The sentence is vague and needs to be rephrased.”

3- “we recorded mfERGs with natural pupil using skin electrodes, and we focused on the nasal responses within the central 15º.” It is better to write why you focus on nasal responses in the introduction too.

4- The mfERG results may be influenced by age, how did the authors consider the effect of age on the mfERG results. Are the groups matched based on age? As you have mentioned, patients with moderate and advanced glaucoma were also significantly older than the normal subjects

5- The authors conclude that “These results indicate that the amplitudes of P1 and P2 in the nasal visual field of the second-order response of the mfERG are a good marker for the stage of glaucoma.” It is better to compare the mentioned parameters ( P1 amplitude,…) between the groups. (between preperimetric and mild, mild and moderate, …)

6- Although the authors showed a significant correlation between the mfERG parameters and MD ,cpRNFL , and mRNFL but I can not find any correlation between the mf ERG parameters and the glaucoma stage.

7- Please summarize the result section, you can omit the parts which have been mentioned in table 1.

8- please explain the p value in table 2

6. PLOS authors have the option to publish the peer review history of their article (what does this mean?). If published, this will include your full peer review and any attached files.

Reviewer #1: No

Reviewer #2: **Yes: **Hamid Riazi-Esfahani

---

## [Author Response · Author response to Decision Letter 0]

21 Aug 2022

We thank the editor and reviewers for their suggestions and comments. These were helpful in allowing us to revise our manuscript to make it better. We have written the revised parts with red fonts in the text and addressed all of the comments below.

I have also edited the Method section to include ethics statement as it was pointed out by the editor. It is also in red fonts. 

Reviewer #1: The authors investigated the role of multifocal electroretinograms to determine stage of glaucoma. They used the amplitudes of the positive peaks (P1, P2) of the second order kernels in the nasal and temporal fields within central 7.5° diameter, reporting that the parameters were significantly different among normal and glaucoma, and glaucoma groups classified by glaucoma severity. Although their study is interesting, it is necessary to suggest the sufficient rationale for assessing nasal side of the macular lesion and nasal to temporal asymmetry in glaucoma.

Glaucomatous damage is characterized by the superior and inferior asymmetry, not nasal and temporal asymmetry, and most frequently affected lesion of glaucoma at the early stage of disease is the temporal side (nasal step defect in early glaucoma.

Answer: We have added the rationale for assessing the nasal side of the macular lesion and nasal to temporal asymmetry in glaucoma in the Introduction section.

And we have added the report of Lee et al in the References.

Reviewer #2: 

This is an interesting study evaluation of the " To determine whether multifocal electroretinograms (mfERGs) recorded with natural pupils and skin electrodes can be used to assess the stage of open angle glaucoma (OAG). "The scientific work done commendable. Overall manuscript is well-written and easy to read with good explanations and interesting findings.

1- There are so many grammatical and syntax errors that need to be addressed.

Answer: This revised copy was edited by native English speaking Professor Emeritus Duco Hamasaki of the Bascom Palmer Eye Institute.

2- However, the pattern ERGs (pERGs) have been shown to managing glaucoma.” The sentence is vague and needs to be rephrased.”

Answer: We have revised this sentence in the Introduction section.

3- “we recorded mfERGs with natural pupil using skin electrodes, and we focused on the nasal responses within the central 15º.” It is better to write why you focus on nasal responses in the introduction too.

Answer: We have added the reason for why we focused on the nasal responses in the Introduction section (same as Reviewer #1).

4- The mfERG results may be influenced by age, how did the authors consider the effect of age on the mfERG results. Are the groups matched based on age? As you have mentioned, patients with moderate and advanced glaucoma were also significantly older than the normal subjects

Answer: We have described the effects of age on P23 (revised manuscript with track changes), and we have added paper by Nabeshima T in the References.

5- The authors conclude that “These results indicate that the amplitudes of P1 and P2 in the nasal visual field of the second-order response of the mfERG are a good marker for the stage of glaucoma.” It is better to compare the mentioned parameters (P1 amplitude,…) between the groups. (between preperimetric and mild, mild and moderate, …)

Answer: We compared the parameters between the groups and described the results in the Results section and Table 3 (new Table).

6- Although the authors showed a significant correlation between the mfERG parameters and MD, cpRNFL, and mRNFL but I can not find any correlation between the mf ERG parameters and the glaucoma stage.

Answer: We have compared the parameters between the groups and listed the Results in Table 3.

On p19-20 (revised manuscript with track changes) of the Discussion section, we also described these findings in the Results section. 

7- Please summarize the result section, you can omit the parts which have been mentioned in table 1.

Answer: We have deleted the parts which have been listed in Table 1. 

8- Please explain the p values in table 2

Answer: We have added an explanation of the P values.

---

## [Decision Letter · Decision Letter 1]

14 Nov 2022

Use of multifocal electroretinograms to determine stage of glaucoma

PONE-D-22-03506R1

Dear Dr,

We’re pleased to inform you that your manuscript has been judged scientifically suitable for publication and will be formally accepted for publication once it meets all outstanding technical requirements.

Kind regards,

Aparna Rao

Academic Editor

PLOS ONE

Reviewers' comments:

Reviewer's Responses to Questions

**Comments to the Author**

1. If the authors have adequately addressed your comments raised in a previous round of review and you feel that this manuscript is now acceptable for publication, you may indicate that here to bypass the “Comments to the Author” section, enter your conflict of interest statement in the “Confidential to Editor” section, and submit your "Accept" recommendation.

Reviewer #2: All comments have been addressed

Reviewer #3: All comments have been addressed

Reviewer #4: All comments have been addressed

2. Is the manuscript technically sound, and do the data support the conclusions?

Reviewer #2: Yes

Reviewer #3: Yes

Reviewer #4: Yes

3. Has the statistical analysis been performed appropriately and rigorously? 

Reviewer #2: Yes

Reviewer #3: Yes

Reviewer #4: Yes

4. Have the authors made all data underlying the findings in their manuscript fully available?

Reviewer #2: Yes

Reviewer #3: Yes

Reviewer #4: Yes

5. Is the manuscript presented in an intelligible fashion and written in standard English?

Reviewer #2: Yes

Reviewer #3: Yes

Reviewer #4: Yes

6. Review Comments to the Author

Reviewer #2: In this manuscript the authors wants to determine whether multifocal electroretinograms (mfERGs) recorded with natural pupils and skin electrodes can be used to assess the stage of open angle glaucoma

(OAG).

The authors addressed most of the comments.

Reviewer #3: The authors have provided adequate responses to all the issues raised by the reviewers. The paper is now ready for publication.

Reviewer #4: The manuscript provides a nice organization and presented well. The revision has been well done by the authors.

7. PLOS authors have the option to publish the peer review history of their article (what does this mean?). If published, this will include your full peer review and any attached files.

Reviewer #2: **Yes: **Hamid Riazi-Esfahani

Reviewer #3: No

Reviewer #4: **Yes: **Takuhei Shoji

---

## [Editor Report · Acceptance letter]

1 Dec 2022

PONE-D-22-03506R1 

Use of multifocal electroretinograms to determine stage of glaucoma 

Dear Dr. Moroto:

I'm pleased to inform you that your manuscript has been deemed suitable for publication in PLOS ONE. Congratulations! Your manuscript is now with our production department. 

Kind regards, 

on behalf of

Dr. Aparna Rao 

Academic Editor

PLOS ONE